# Sediment Budgets for Small Salinized Agricultural Catchments in Southwest Australia and Implications for Phosphorus Transport

**Robert J. Wasson** [1,2,*] **and David M. Weaver** [3]

1  College of Science and Engineering, James Cook University, Cairns, QLD 4878, Australia
2  Fenner School of Environment and Society, Australian National University, Canberra, ACT 2601, Australia
3  Department of Primary Industries and Regional Development, 444 Albany Hwy, Albany, WA 6330, Australia; david.weaver@dpird.wa.gov.au
*  Correspondence: robert.wasson@jcu.edu.au

**Abstract:** Examples of sediment budgets are needed to document the range of budget types and their controls. Sediment budgets for three small agricultural catchments (7.6 to 15.6 km$^2$) in southwestern Australia are dominated by channel and gully erosion, with sheet and rill erosion playing a subordinate role. Erosion was increased by clearing naturally swampy valley floors and hillslopes for agriculture and grazing, and episodic intense rainstorms. The proportion of sediment from channel and gully erosion in the sediment budget appears to be determined by the depth of alluvial fills. Dryland salinization caused by clearing native vegetation has connected hillslopes to channels across narrow floodplains, increasing the Sediment Delivery Ratio (SDR). Yield and SDR are found to be insensitive to major in-catchment changes of vegetation cover after initial clearing, the ratio of sheet and rill erosion/channel and gully erosion, and sediment storage masses. This supports the idea that yield alone is often a poor indicator of the impact of land use and land management change. Riparian vegetation would reduce sediment yield but not phosphorus yield. This study demonstrates the value of mixed methods where field observations and chemical analysis are combined with information from local people.

**Keywords:** sediment budget; dryland salinization; sediment delivery ratio; Western Australia; Kalgan River; phosphorus

## 1. Introduction

Sedimentation and estuarine eutrophication have been identified as serious problems globally. Sediment and phosphorus (P) transport are often strongly linked [1] and focus attention on the management of particulate P. Recent studies in sandy catchments identify soluble P as the dominant P form being transported [2–5], bringing into question the use of practices that reduce P transport through sediment control. Soluble P is often the dominant form of P in sandy catchments because phosphate is not removed from the solution by adsorption processes [2].

The Kalgan River catchment [6–8], along with other rivers in southwestern Western Australia, suffers from issues associated with sediment and P transport. Water quality measurements at different scales in the Kalgan River catchment suggest unconnected sources of sediment and P [2]. The Kalgan River catchment is also a place where the conjunction of land-use/land cover change, salinization, erosion, and loss of P can be investigated together to contribute to our understanding of material budget changes.

It has been argued that the major scientific challenges in the study of material budgets in river catchments are to discover how common various types are, define the major controls on these types, and determine the characteristic time periods for each of them [9]. The Kalgan River catchment, with its conjunction of problems, presents an opportunity to add to our knowledge of sediment budget types [9] and make some progress in defining the major controls and temporal trajectories of change. The study catchments examined here present

an opportunity to explore a role in a sediment budget for dryland salinization, a process that follows the clearing of native vegetation and also leads to the death of the remaining vegetation or that which grows on parts of the cleared land, thereby reducing resistance to soil erosion. To our knowledge, this is the first study that includes salinization as a causal process in a sediment budget, although others [10] make relevant observations and note a role for salinization in erosion [11]. Whilst most of this paper is concerned with sediment budgets, some data relevant to a P budget are also presented. To that end, sediment budgets are constructed for three small sub-catchments of the Kalgan River catchment in southwestern Western Australia, mainly for the period since European settlement.

## 2. Materials and Methods

### 2.1. The Kalgan Catchment Environment

The catchment has an area of 3041 km$^2$ and drains into an estuary called Oyster Harbour, and on its shore lies the town of Albany. The major river is the Kalgan, recorded as Kalgan-up by Alexander Collie in 1831 (History of River Names, http://www0.landgate.wa.gov.au/maps-and-imagery/wa-geographic-names/name-history/History-of-river-names#K; accessed on 23 August 2021). The catchment rises in the Stirling Range (consisting of Proterozoic sandstone and quartzite), 70 km north of Albany. The landscape is undulating, and is cut into Tertiary Pallinup Siltstone (spongolite, siltstone, sandstone, and minor limestone). These soft rocks are overlain by sand, of both residual and aeolian origin, some of which is lateritized [12]. Outcrops of Proterozoic granite and adamellite form the Porongorup Hills, 35 km north of Albany, and form small hills in the catchment, and, along with Proterozoic gneiss, forms the basement to the Tertiary sedimentary rocks. The unconformity between the Proterozoic and Tertiary rocks is an uneven surface created by erosion, uplift, and faulting. The Werillup Formation of siltstone, spongolite, and gravel overlies the unconformity, and is a preferred pathway for highly saline groundwater (>10,000 mg/L TDS) [13].

Summer is temperate and dry, while winter is cool and wet. Annual average rainfall ranges from 400 mm in the northeast of the catchment to 900 mm near the coast. The natural vegetation, according to the first European explorers in the early to mid-nineteenth century, consisted of tall, open (and some closed), eucalypt forest in the west and south, grasslands south of the Stirling Range, and, in some parts of the plains near the Kalgan River, mallee (multi-stemmed) eucalypt woodlands, and tall shrublands with grassy plains in the eastern area. The riparian zone along the Kalgan River varied from south to north from a closed, tall, eucalypt forest with paperbarks (Melaleuca spp.), Trimalium sp., Hakea spp. and Banksia spp., rushes, sedges, ferns, and grasses to low closed forests with Melaleuca spp., rushes and samphires [6].

By 1991, 66% of the upper catchment and 88% of the lower catchment was cleared for cropping and grazing [6]. The soils are mostly shallow, grey, acidic, siliceous sands overlying laterite high in the landscape, and sands, sandy gravels, and limited areas of sandy loams in valley floors [14].

Once cleared, agriculture was only possible with the use of phosphate fertilizers, usually applied annually [15]. Eutrophication of Oyster Harbour, and other coastal estuaries and lagoons, has stimulated research into P and sediment budgets for rivers in this part of Western Australia [16]. Sedimentation also reduced the volume of river pools, compromising aquatic habitats and speeding flood flows. Erosion of valley floors increased the opportunities for saline groundwater to reach streams, also degrading aquatic habitats and increasing stream salinity. At the large catchment scale, about 40% of P measured in rivers is dissolved, and 60% is in particulate form [17], whilst at the hillslope scale 96–99% of measured P loss is dissolved [2], suggesting dissolved P is converted to particulate P in streams [18,19]. Most of the dissolved P lost from agriculture is derived from what are now overly P-fertile soils due to long-term application of phosphatic fertilizers [2,15,20].

Groundwater is highly saline (from cyclic salts), and water tables continue to rise due to reduced evapotranspiration from land clearing, bringing inherently saline water closer to plant roots [13,21]. Aquifers are local and are controlled by bedrock, including the

unconformity at the top of the Proterozoic crystalline basement. Near the northern and eastern margin of the Kalgan catchment, a low scarp was cut into the Pallinup Siltstone as the catchment has cut headward since the Eocene. Natural brackish seepage occurs at the base of this scarp, often along the unconformity, and allowed the development of samphire swamps. Since clearing native vegetation for agriculture, saline seepages increased in both number and discharge. Recharge in the cleared sandplain to the east of the integrated drainage area of the Kalgan catchment is feeding the seepages, which are expected to get worse in coming years. The samphire swamps are becoming more saline, and marginal vegetation is disappearing.

### 2.2. The study Subcatchments

Three small subcatchments on the eastern side of the Kalgan catchment were chosen, rising on the low escarpment cut in the Tertiary cover rocks in the Kamballup–Takalarup area. Details of the three sites are provided in Table 1. The area has an approximate mean annual rainfall of 600 mm and is considered part of the Upper Kalgan catchment [6] where salinization is common.

**Table 1.** Details of the three study catchments.

| Name | Area (km$^2$) | Relative Relief (m) | Strahler Stream Order | Drainage Density (km/km$^2$) | Average Gradient | |
|---|---|---|---|---|---|---|
| | | | | | Channelized Part | Whole |
| Takalarup Creek | 15.6 | 70 | 3 | 0.7 | 0.0162 | 0.0170 |
| Dingo Creek | 7.6 | 60 | 2 | 0.6 | 0.0256 | 0.0217 |
| Salt Creek | 14.3 | 60 | 3 | 1.2 | 0.0129 | 0.0124 |

The study sites were selected to be as comparable as possible with those investigated in southeastern Australia [22–24], where sediment budgets are dominated by channel and gully erosion. The southeastern Australian sites were affected by land use change in the latter part of the first half of the nineteenth century and the early 1900s. The Kalgan River study catchments were cleared during the last 100 years, and, in some cases, the last 50–60 years. As a result, the early stages of the processes of geomorphic change triggered by land use change can be detected, and many of the original (or near original) farmers are still alive to provide information. This opportunity for documentation of the early stages of change is not available in southeastern Australia.

The study area is first explicitly mentioned in the reports of European explorers in December 1831, when surveyor Raphael Clint described 'tolerable' soil near a stream that Aboriginal people called Ta-kel-e-up (Takalarup), where the water was brackish [25]. Brackish and highly saline water was also found in pools in the Kalgan River at Coon-bil-up (Kamballup) nearby. From experiences elsewhere in the catchment, early summer stream salinity followed poor winter rainfall. Presumably, saline groundwater inflow to the streams were not diluted by surface runoff. The clearing of native vegetation for agriculture and pastoralism reduced evapotranspiration, leading to outcrop of the saline water table and, therefore, surface salinization. This leads to the death of remaining vegetation and resistance to soil erosion is reduced.

By the 1850s, sheep were grazing the riverbanks, and probably the valley floors of tributaries. A stone hut was built at this time at Takalarup, and permanent settlement was established by the 1870s for grazing purposes [25]. According to the local historian Joan Blight (pers comm.), the land was used for grazing by John Hassell between 1890 and 1900 on the Kalgan frontage just upstream of Takalarup.

Each of the three subcatchments in the Takalarup area were described, a sediment budget was derived, and a history was documented. Of these three creeks (Takalarup Creek, Dingo Creek and Salt Creek), most detail is presented for the Takalarup Creek subcatchment.

### 2.3. Takalarup Creek Catchment—Land Use, Erosion, and Salinization History

As many of the original (or near original) farmers in the study catchments were still alive to provide information, oral histories were collected. These were supplemented with information from historical aerial photography and survey plans, interpolated climatic data (http://www.longpaddock.qld.gov.au/silo/ accessed on 27 January 2021), and other physical and chemical measurements in this study to support interpretation of sediment and P data. For comparison with the Takalarup Creek history, a gully called Moorialup Creek ~12 km SW of the Takalarup Creek–Kalgan River junction was also examined, as was Noorabup Creek, which is 4 km to the south of Takalarup. In addition, gullying in a creek 16 km west of Redmond was also examined and compared with other sites (Figure 1).

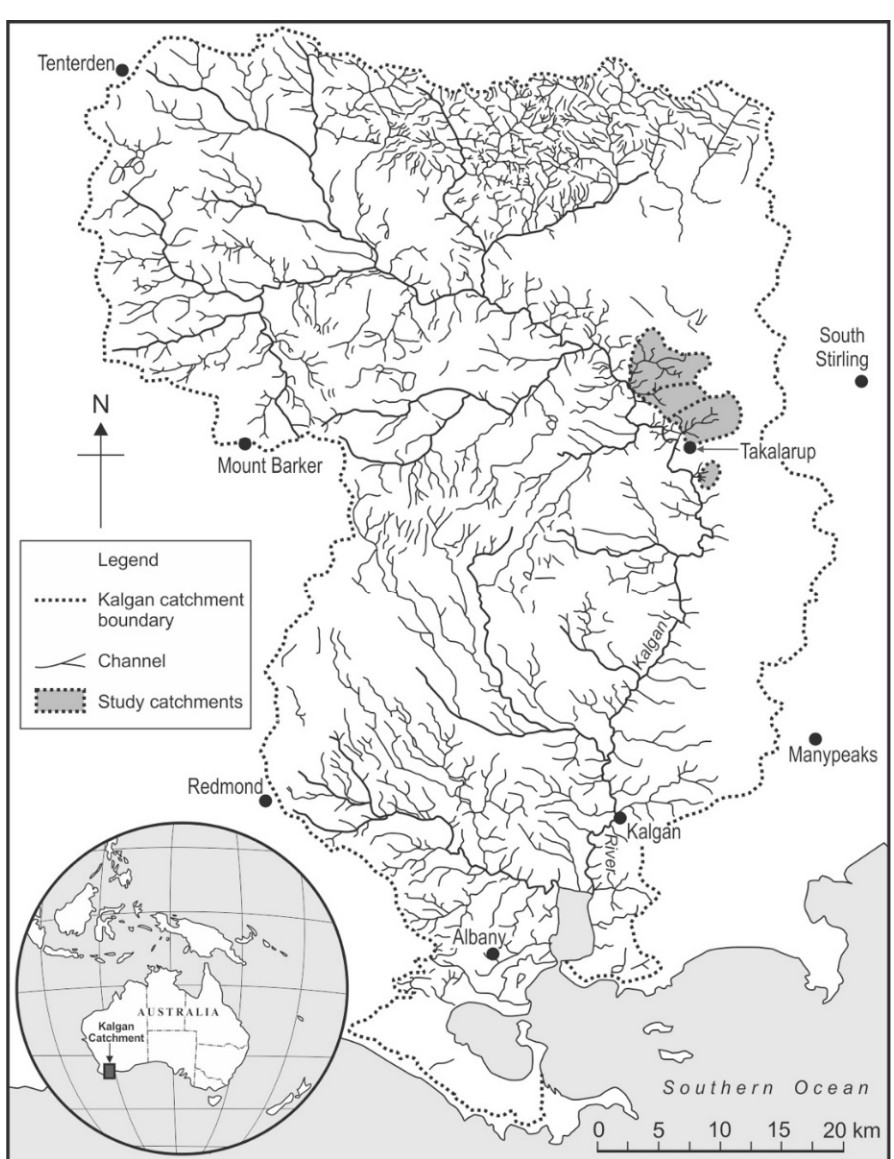

**Figure 1.** Kalgan catchment, stream network, and study catchment locations.

### 2.4. Takalarup Creek Catchment—Sediment Budget

Sheet and rill erosion were estimated from data collected from farm dams in the Kalgan catchment [26]. Twenty-four dams with catchments ranging in size from 0.9 to 43.8 ha and with average gradients between 0.5° and 6.2° (ranging from 7 to 45 years in age; either when built or last cleaned out) were examined in the Kalgan catchment. The volume of sediment in each was determined by probing from a boat, and by coring to

ensure that the probing was accurate. Because only the mineral fraction is of interest, the volumes of mineral sediment were corrected for organic content by subtracting loss on ignition values. The volume, then mass after correction for bulk density, was also corrected for the material derived from dam walls by stock trampling, wave attack, and sheet and rill erosion by estimating the volume lost below a projected surface between those parts of the dam walls which were not disturbed.

Catchment soil loss was calculated from the corrected dam sediment volume alone because the dams either do not overflow or lose very little water and sediment. Trap efficiency calculations for the dams were not performed and given that the clay content is generally <10% in the catchment soils, loss of this fraction by overflow will have little effect on the calculated catchment sediment yield. Catchment sediment yield varied from 20 to 310 t/km$^2$/year.

Sediment yield generated by sheet and rill erosion from hillslope segments was estimated by using a regression relationship from measurements in the farm dams, as follows:

$$y = 1.0717 \times X^{0.48124}; n = 24; r^2 = 0.44 \ (p > 95\%) \tag{1}$$

where $y$ is sediment yield (t/year) and $X$ is catchment area (ha). The catchments used in this relationship are no larger than the subcatchment hillslope segments, thereby avoiding scale-dependent phenomena. Also, erosion in these small catchments had a small effect on yield which is overwhelmingly dominated by sheet and rill erosion [26]. There are small gullies in a few of the farm dam catchments but their impact on the total sediment yield is very small. Hence the contribution to the dams is almost all from sheet and rill erosion.

For the three subcatchments under investigation here, hillslope segments were defined by first subdividing all channels in the study catchment into 100 m reaches. At each 100 m point, orthogonals and contours were drawn to the catchment divides. Where hillslope segments are separated from channels by a low gradient valley fill, it is assumed that the yield from those segments did not reach the channel network, except where salt scalds (shallow areas eroded after saline water discharge has killed vegetation) expanded from the valley fill surface to the adjacent foot slope. In these cases, part of the eroded soil washed into channels, and some was deposited on the valley fill. The deposited mass was not subtracted from the calculated yield from the adjacent hillslope segment because it is clearly highly mobile and will soon reach the channel. The volume (and mass) of the scalds was measured and added to the calculated adjacent hillslope segment yield.

The sheet and rill erosion yields were calculated for current conditions, given that Equation (1) is based mainly on data for recent years. To estimate this term of the sediment budget for previous periods where vegetation had not been cleared or was only partially cleared (based on aerial photographs), the SOILOSS equation [27] was first used. SOILOSS is a version of the RUSLE, adapted for Australia by using locally measured data. The relationship between estimates derived from SOILOSS ($y$, t/year) and the regression relationship based on data from farm dams ($X$, t/year) for modern conditions is:

$$y = 1.1927 \times X^{0.19196}; n = 24; r^2 = 0.93 \ (p > 95\%) \tag{2}$$

using a range of representative locations. This relationship was then used to adjust the estimates from SOILOSS for periods when vegetation cover was greater than now. This approach was taken because of the inapplicability of SOILOSS to long slopes [28].

To test the calculated proportion of channel erosion products in the sediment actively being transported in the modern channel, $^{137}$Cs in the <10 μm fraction was measured by high resolution gamma spectroscopy. This can be used to estimate the proportion of topsoil in the river sediment and thus, by difference, the proportion from subsoil erosion by channel and gully incision [29].

The same calculations were used to estimate the sediment budget for the period of 1955–1997. The area of clearing was taken from the 1990 aerial photographs. Because of the uncertainties associated with dating the Post Settlement Alluvium (PSA) (other than

it is post-European and mostly post-1939) and the fan deposits (other than they are post-European and mostly deposited between 1939 and 1955), a budget for the entire period 1903–1997 is also presented.

The total PSA was estimated by reconstructing the amount removed by further incision after 1955, minus that deposited in the downstream reach after 1955. The main channel and connected gully erosion mass was calculated by assuming that two-thirds occurred in 1939 (based on the observations of Laurel and Ian Lock), and that the gullies were graded to the main channel. Yield was calculated as the difference between the total of erosion products delivered to the channel minus storage, as discussed earlier.

Main channel erosion was calculated by measuring the volume (corrected for measured bulk density) of the incision below the top of the pre-PSA valley fill. Volumes and masses of gully erosion were measured by field survey, using PSA, original valley floor form, and the surface of the A horizon exposed in gully walls to guide the measurements.

Colluvial sinks were not estimated as there is no clear distinction in the field of post-agricultural deposition on hillslopes. The fan at the lower end of the catchment was surveyed and depth was determined by digging pits and by auger. The base of the fan deposits is easily identified overlying organic-rich sediment. The PSA mass was estimated by field survey and augering, with measurement of bulk density.

The yield of the catchment before 1903, when grazing by sheep of native pasture in an uncleared catchment was the only land use, was likely to be very low and was estimated by assuming that the valley fill, in the downstream reach, represents yield and trapped either 100% (so no sediment reached the Kalgan River) or 50% of the sediment delivered to it. These pre-clearing estimates are not unreasonable given comparisons between pastured and plantation-forested catchments show a 2.5 times greater sediment load for the former than the latter [30], whilst other research suggests specific sediment yield following logging increases an order of magnitude compared to a control [31]. For the period after 1903, yield was estimated by the difference between erosion of channels and gullies, plus the quantity of the products of sheet and rill erosion delivered to the channel network, plus eroded PSA, minus PSA, and fan deposition. This method clearly leads to uncertain estimates, given the errors in the various terms of the sediment budget. Therefore, yield was checked for the most recent period against measured suspended sediment loads in the Kalgan catchment. Measured suspended sediment loads from 26 catchments of varying size were available from the WA Department of Primary Industries and Regional Development, including subcatchment areas of 0.95 to 2427 km$^2$ and from 1 to 10 years. The regression equation is:

$$y = 5.7772 \times X^{0.7181}; n = 26; \ r^2 = 0.44 \ (p > 95\%) \tag{3}$$

where $y$ is the mean measured suspended sediment load in t/year, and $X$ is the area of each measured subcatchment in km$^2$.

*2.5. Dingo and Salt Creek—Sediment Budgets*

Sediment budgets for Dingo and Salt Creek catchments (Figure 2) were constructed using the methods applied to the Takalarup Creek catchment. Incision volumes were estimated using the base of the PSA and/or the surface of remnants of pre-incision floors as the datum.

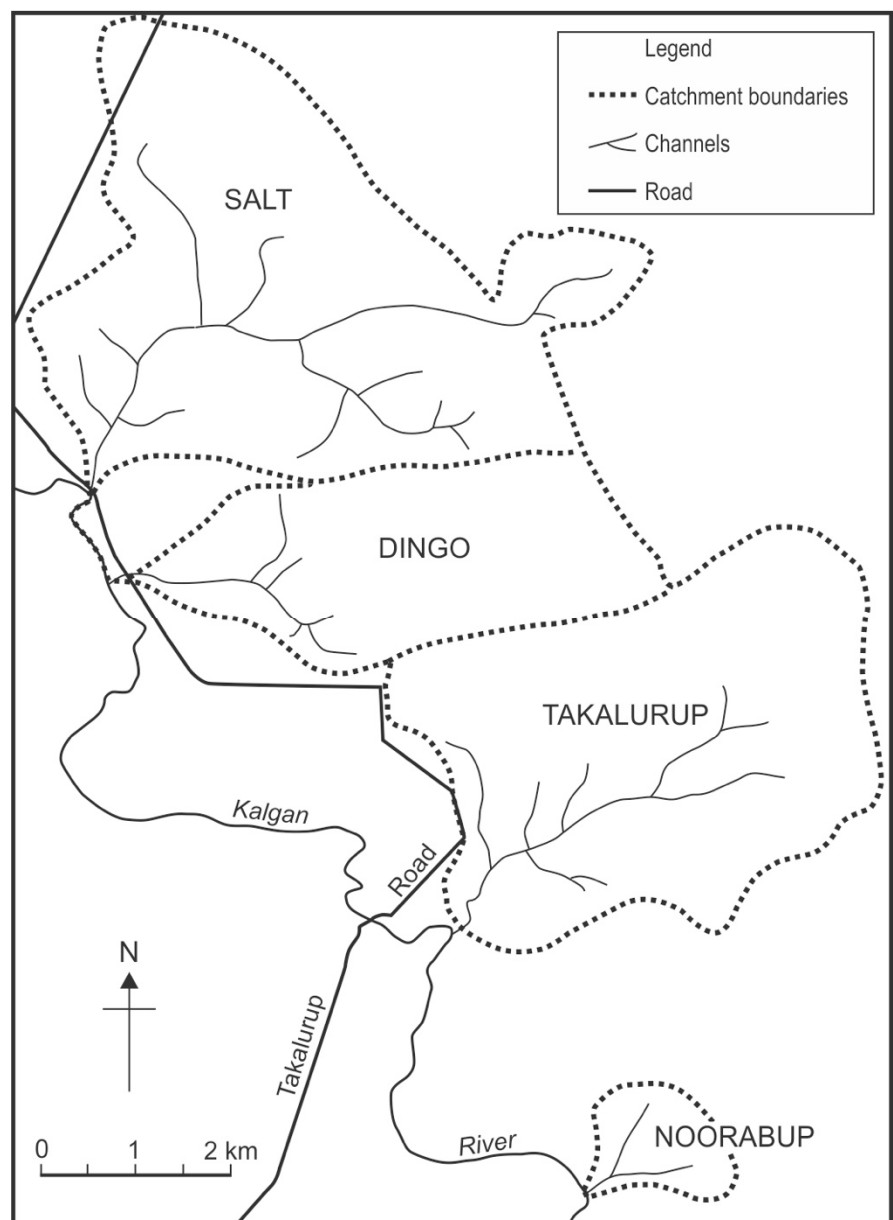

**Figure 2.** The study catchments.

### 2.6. Takalarup Catchment Phosphorus Loss

Soil samples of swampy meadow and alluvial fill from profile 4 (Figures 3 and 4), and of tributary sediment and the main channel were collected. Each was separated into <75 μm and >75 μm fractions, air dried, and analyzed for Total P (TP) using Kjeldahl digestion [32], bicarbonate extractable (plant available) P (P(HCO$_3$)) [33], and biologically available P (P(Bio)). P(Bio) was determined by adsorption onto an iron oxide impregnated filter paper strip in a 0.01 M CaCl$_2$ solution [34]. The P in extracts and digests was measured colorimetrically as the phosphomolybdenum blue complex [35].

A regression equation was developed from measured P load data from 36 catchments of varying size from the Department of Primary Industries and Regional Development, over subcatchment areas of 0.95 to 2427 km$^2$ and 1 to 10 years [2]. The equation from these data is:

$$y = 1.3137 \times X^{0.7189}; n = 36; r^2 = 0.52 \ (p > 95\%) \tag{4}$$

where $y$ is P load (kg) and $X$ is area (ha) was used to compare to estimated P yields for the Takalarup Creek catchment determined from the sediment budget and soil P analysis.

## 3. Results

### 3.1. Takalarup Creek Catchment—Land Use, Erosion, and Salinization History

A survey by Hugh Russel conducted in June 1903 of part of the lower reaches of the catchment noted 'good agricultural land' along the main creek with yate gums (*Eucalyptus occidentalis*). On the hillsides and hilltops, dense mallee (multi-stemmed eucalypts), low scrub, and ironstone gravel are noted. About 750 m upstream from the junction with the Kalgan River, close to the left bank of the creek, the surveyor noted 'now being cleared and ploughed'. Nearby, 'good loamy soil lightly timbered with yate gums', and 'good black soil' near the creek. Once again, mallee and gravel are noted on hillslopes. About 1.5 km upstream from the Kalgan River, Russell noted 'good loamy soil' on the left bank of the main creek, 'fair soil on the slope' and 'fair gritty soil' on a hilltop, and once again 'few yate gums', 'dense mallee', and 'ironstone gravel'.

Interestingly, the outlines of the main creek and tributaries are shown by the surveyor, suggesting a defined channel in 1903. The creek is labelled 'Takalarup Gully' on the hand-drawn chart, but on the final drafted version is called 'Takalarup Creek'. Given that the word "gully" at that time was used to refer to many kinds of channels, no inference about the cross-sectional shape of the channel can be drawn (B. Starr, pers. comm.). In addition, the plan of 1903 shows Takalarup Creek joining the Kalgan River as a defined channel. This is no longer the case, with an alluvial fan intervening between the defined tributary channel and the river.

According to Ian Lock, who lived at Takalarup from 1950 on, the valley floor (to ~1500 m from the river) was cleared between 1910 and 1920 judging from the condition of pasture, style of fencing, and absence of tree stumps when he arrived. According to Ian Lock, most of the land of the catchment away from the main creek was cleared after the World War II—a conclusion confirmed by examination of aerial photographs.

According to both Ian and Laurel Lock, who were interviewed separately, the cleared valley-floor land was used for the cultivation of potatoes and cereals and for grazing between 1910 and 1955. Sheep grazed the uncleared land as well. In 1939, an intense rainstorm (43 mm on 20 Jan and 88 mm on 21 Jan, based on data extracted from interpolated climatic data (http://www.longpaddock.qld.gov.au/silo/; accessed on 23 August 2021) eroded the main valley floor and the left bank tributary ~650 m upstream of the Kalgan River, according to Laurel Lock who moved to Takalarup in 1949 and left in 1977. Information about the storm came from both Mr. H Gibbons of Moorialup and Mr. Jack Rowe, a former owner of Takalarup farm. The same storm eroded gullies at other sites in the district, according to Ian Lock.

Salt scalds existed in 1949 along the margins of the main creek, according to Laurel Lock. Both Ian and Laurel Lock believe that the salinization resulted from removal of paperbark trees (*Melaleuca cuticularis*). The scalds were treated with straw, and they have not spread since the 1950s. Further upstream, where salt scalds are now common, salinization was not serious until 1977.

Ian and Laurel Lock recall that the main creek deepened by about a third since the mid 1950s. Laurel Lock has a vivid memory of major erosion of the main creek in 1955 which 'silted up' the creek near its junction with the Kalgan River. The channel (presumably that shown on the 1903 survey plan) was buried, then reformed in a different location, then shifted back and forth across what is now an alluvial fan. Ian Lock confirmed the broad features of Laurel Lock's recollections. Based on data extracted from interpolated climatic data (http://www.longpaddock.qld.gov.au/silo/; accessed on 23 August 2021) the Takalarup Creek catchment experienced 24 mm of rain on 16 February, 77 mm on 17 February, and 22 mm on 18 February 1955.

Survey plans, recollections of local people, and the 1943 and subsequent aerial photographs show that most of the catchment was cleared by 1955. Grazing with some improved pasture and cropping for cereals were established by then and continue to the present.

The lower 1500 m of the main valley floor of the Takalarup catchment consists of an alluvial fill with colluvium interbedded at the margins. The stratigraphy of the fill is shown in Figures 3–5 at 10 of the cross sections located on Figure 3.

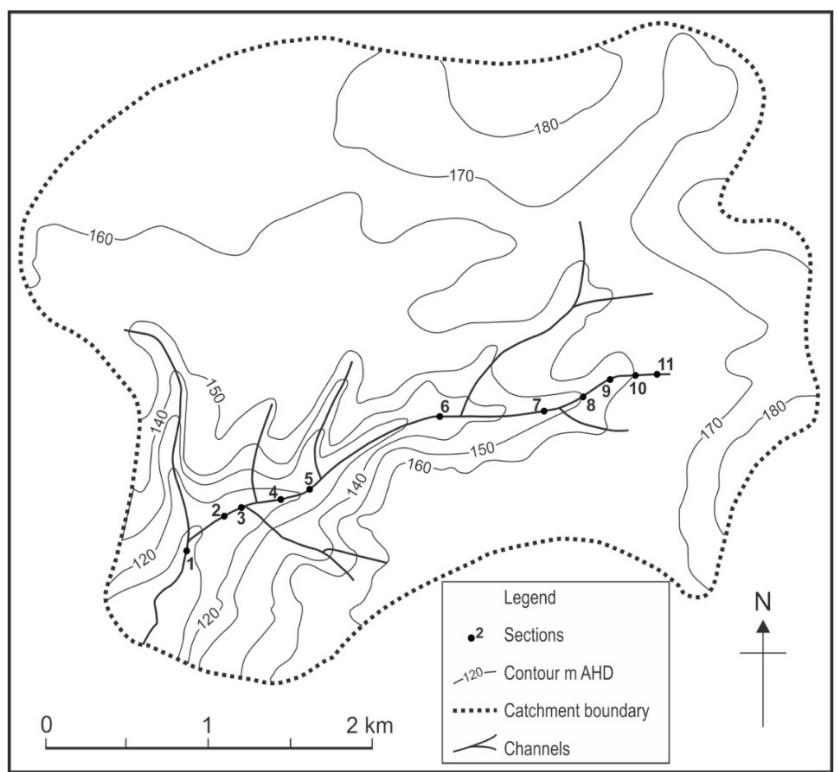

**Figure 3.** Takalarup Creek catchment and section locations. Numbers on the figure are cross sections and sampling sites.

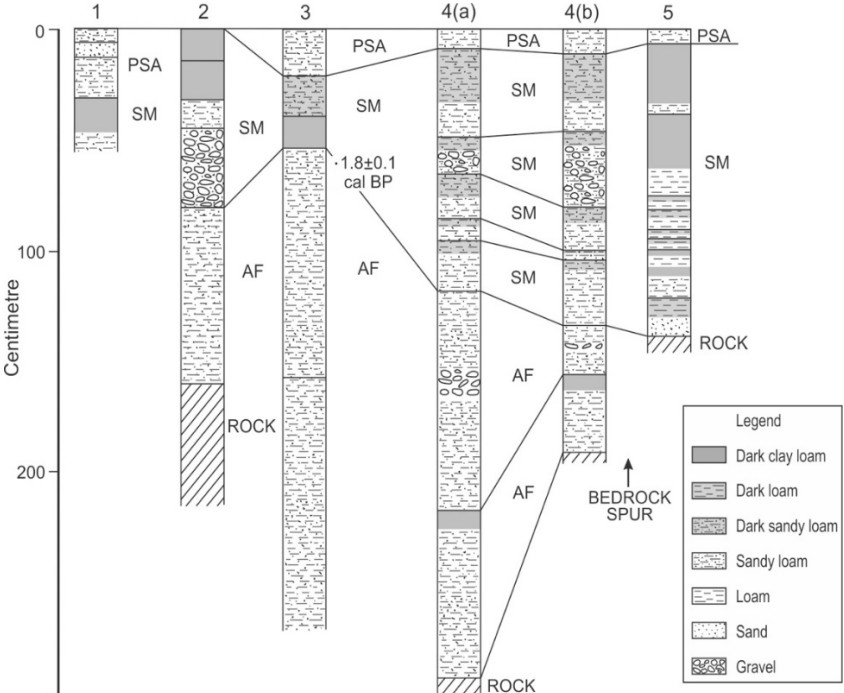

**Figure 4.** Stratigraphic sections **1–5** of Takalarup Creek. PSA = Post Settlement Alluvium, AF = Alluvial Fill, SM = Swampy Meadow.

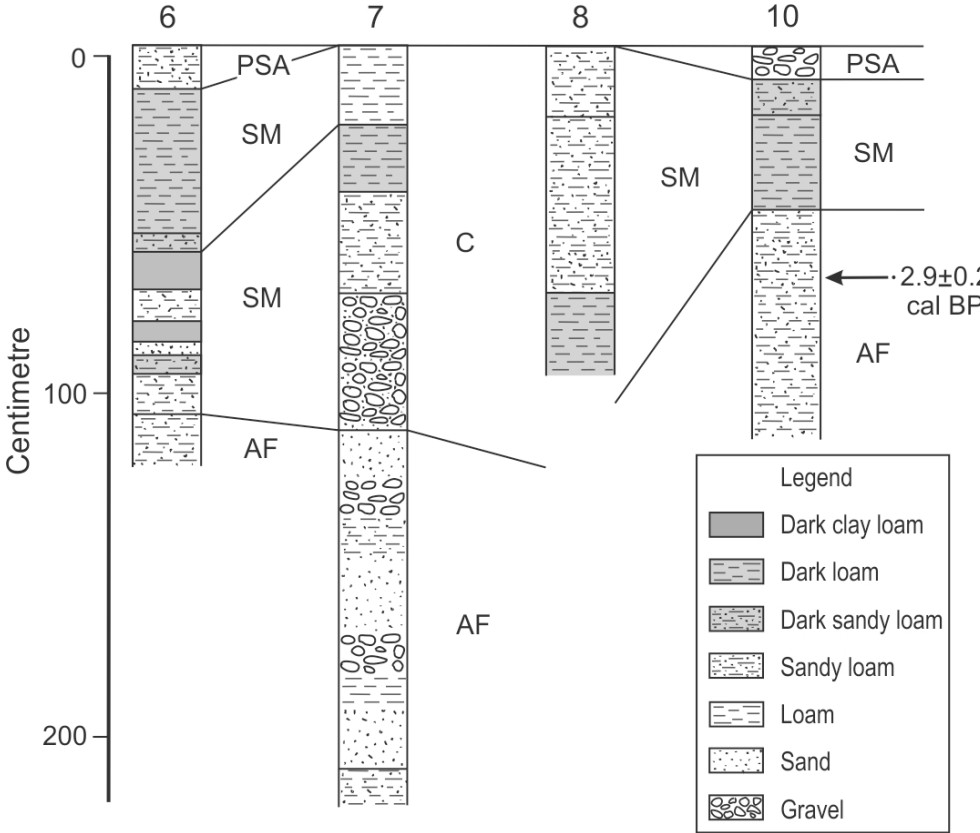

**Figure 5.** Stratigraphic sections **6–10** of Takalarup Creek. PSA = Post Settlement Alluvium, AF = Alluvial Fill, SM = Swampy Meadow, C = Colluvium.

At nine of the cross sections (1, 3, 4(a) and 4(b) which are 23 m apart either side of 4, and cross sections 5, 6, 7, 8, 10) the top of the profile consists of a pale brown to yellow brown laminated sandy loam and fine sand with gravel in cross section 10 (Figure 3). At cross sections 1 and 3, this layer contains the bones of sheep and pieces of fencing wire, showing that it is of post-European age. It is equivalent to the PSA described from southeastern Australia [23] and the U.S.A. [36].

Beneath the PSA is a black and dark brown clay loam or light clay that is found in unincised swampy valley floors elsewhere in the district. At these unincised sites, there is a shallow, ill-defined channel that passes through an organic-rich, fine-grained deposit covered by rushes (particularly Juncus kraussii) and sedges. Similar to southeastern Australia, this landscape element has been called a 'swampy meadow' (SM). The SM consists of several beds of dark, clay-rich sediment with interbeds of sandy loam, fine sand, and in a few cases, fine gravel. This sequence is interpreted as the result of a shallow channel that deposits sand which is then buried by fine sediment from overbank flows when the channel shifts location. Once again, the same sequence was seen at many sites in southeastern Australia.

In the field, the dark layers in the SM deposits appear to have more organic matter than the coarser-grained channel deposits. Loss on ignition (LOI) measurements from the profile at cross section 4(a) shows that the picture is not this simple (Figure 6). The highest LOI is in the PSA and in a channel deposit between 87 and 95 cm. The colour is therefore likely to be the result of organic matter, clay, and possibly Mn.

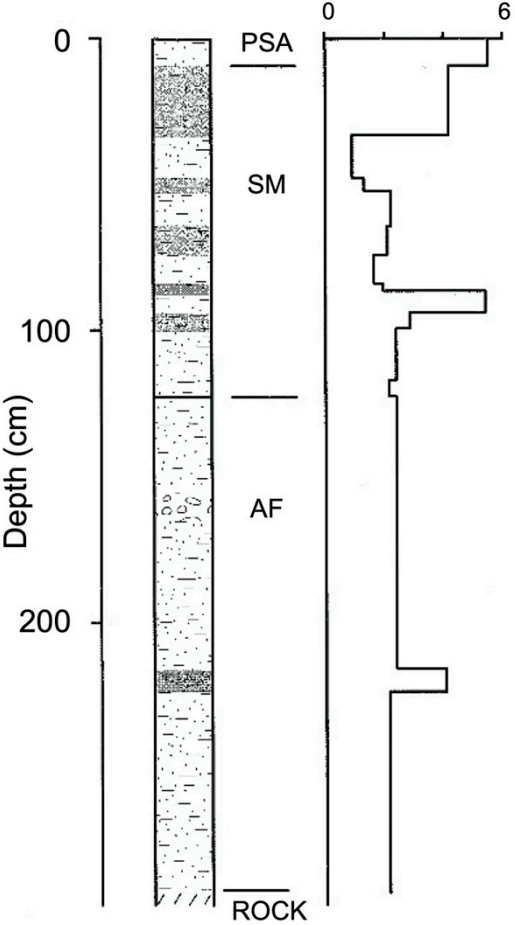

**Figure 6.** Stratigraphic profile and Loss on Ignition (%) at cross-section 4 on Takalarup Creek. PSA = Post Settlement Alluvium, AF = Alluvial Fill, SM = Swampy Meadow. The vertical axis is in centimetres.

The channels in the SM were no deeper than the maximum thickness of the fine-grained units that would have formed their banks. The maximum channel depth was therefore 24 cm, like the intact unincised channels in the district.

In the study, area clearing was underway by 1903 and was complete by 1955. While hillslope erosion can be calculated for the two post-1903 periods, channel erosion is more difficult to estimate. However, most of the incision occurred in 1939, and the farmers Laurel and Ian Lock independently observed that the main channel downstream of the constriction (and therefore the lower parts of tributaries now graded to the main channel), had deepened by only about one-third since the 1950s. The most likely sequence of events is as follows: major incision in 1939 with deposition of the PSA and about one-third of the fan at the lower end of the catchment (assuming the fan volume to be proportional to the volume fraction excavated by channel incision); further incision of the channels, erosion of some of the PSA, and deposition of the rest of the fan during the period between 1955 and 1997.

Gullies are mostly connected to the main channel and are graded to its bed. Therefore, they are also post-European, the additional evidence being PSA at their downstream reaches. In many cases the pre-gully valley floor shape can be seen where a gully has formed to the side of the original drainage line.

At cross section 3, detrital charcoal was dated by radiocarbon to 1890 + 140 BP (CS309) (1831 ± 169 cal BP, where BP is Before Present, i.e., before 1950 CE) just below the base of the SM. This indicates recent (Late Holocene) development of this landscape type.

Below the SM is an alluvial fill of different character. Brown to pale brown sandy loam, sandy clay loam, gravel, and sands are interbedded. Very little organic matter is preserved in these deposits, and only at cross section 4(a) is there evidence of a deposit like the organic-rich layers in the SM. These alluvial deposits (AF on Figures 4 and 5) appear to have been deposited by braided channels, possibly rapidly, although the chronological support for this idea has not been established. One radiocarbon date of 2750 ± 50 BP (CS311) (2911 ± 179 cal BP) for detrital charcoal from near the top of the AF at cross section 10 indicates that the valley fill may all be of Holocene age.

At cross section 7 (Figure 3) the incised channel of Takalarup Creek was cut into the colluvial footslope on the left bank. Colluvium makes up most of the profile, with layers (from the top down) of loam, clay loam with quartz pebbles, sandy loam with pebbles, and gritty to sandy loam. This section of mostly fine-grained sediment lies on top of more than 1 m of alluvium (AF equivalent) consisting of interbedded fine sand, pebbly fine sand, sandy loam, and pebbles and cobbles. The surface colluvial layer (0–23 cm) can be traced laterally to show that it is coeval with the uppermost organic-rich part of the SM at cross section 6. The remainder of the colluvium appears to be coeval with the basal part of SM at cross section 6. Therefore, the colluvium began to accumulate at the same time as the SM, ~1800 cal BP (~1.8 ka cal BP). The underlying AF at cross section 7 provides evidence of the highest energy conditions in any of the exposed sections, namely layers of fluvial pebbles and cobbles.

The sections exposed along Takalarup Creek indicate higher stream power before ~1.8 ka BP, as evidenced by fluvial gravels, and lower stream power after this time. The last 1.8 ka cal BP are characterized by well-vegetated valley floors, shallow channels, and accumulation of colluvium. The change of conditions ~1.8 ka BP could be the result of climate change, although independent evidence of this is not forthcoming from the area [37].

The shallow channel within the SM was incised to create a channel up to 12 times deeper, beginning in 1939. Deposition of some of the products of incision produced the PSA, probably during the early stages of incision when it was discontinuous and small fans at the mouth of the discontinuous pieces of the channel were formed (cf., [23]). The result is the observed patches of PSA stranded on the bank tops of the incision.

Between cross sections 4 and 5 (Figure 4) the valley floor is narrowed by bedrock spurs of layered gneiss on either side of the incised alluvial fill. The sedimentary units of SM and AF at cross section 5, immediately upstream of the constriction, dip downstream at 6–10°, while between the spurs the dip is up to 3°. The changes of dip and the valley form suggests a fault or flexure on the upstream side of the spurs, which was active in the Late Holocene. Impregnation of the sediments at cross section 5 by iron oxides, and a pH of 10 for the sediments, suggests that saline water, Fe, and probably $Na_2CO_3$ are moving upwards at the fault or flexure. The fault or flexure trends northwest-southeast, approximately parallel to the major fault system that extends northwest from Granite Hill near the eastern end of the Porongorup Hills [12]. The downthrown side is to the northeast, that is, the upstream side of the spurs.

The Moorialup Creek was drained by a shallow depression covered by rushes before the catchment was cleared in 1957/58 by Dudley Wise. Mr. Wise saw this shallow channel incise to a depth of ~3 m during storms in 1982 (62 mm on 21 January and 101 mm on 22 January) and 1983 (18 mm on 22 June) (based on data extracted from interpolated climatic data (http://www.longpaddock.qld.gov.au/silo/; accessed on 23 August 2021). Layered alluvium is exposed in the walls of the gully, with bedded sands and gravels, clays, and sandy clays. At the top is 10–20 cm of dark brown to black loam grading down to sandy loam which forms the surface of the adjacent hillslopes. Patches of PSA up to 10 cm thick occur along the edges of the gully in the downstream reaches. Charcoal, probably from an in situ fire, bedded immediately beneath the dark brown to black cap was dated to 1290 ± 30 BP (CS313) (1190 ± 122 cal BP). This sequence, and the chronology, is consistent

with that described at Takalarup Creek, with a lower energy sediment transport system established ~1.2 ka cal BP followed by gullying ~25 years after clearing.

Two other sites provide evidence of gullying after clearing. On a grazing property ~16 km west of Redmond, a <3 m deep gully formed during an overnight rainstorm in 1977. The gully has cut into deep white sand with an organic A horizon and minimal B horizon development. Clearing of this area occurred in 1967, according to Peter Buxton, a local farmer and former soil conservation officer from Victoria. Another gully further east was produced at the same time.

About 4 km south of Takalarup Creek, Ian Lock and Jack Rowe recall the incision of Noorabup Creek after clearing in the late 1960s, although Jack Rowe remembers some clearing a few years earlier. A shallow, dish-shaped drainage line joined several pools that were a little over 1 m deep. These pools were used as a freshwater source before clearing, with the water being carted away. They are surrounded by paperbark trees (Melaleuca sp.) and rushes. One of the waterholes is still there but is now saline. An incision of ~250 $m^3$ occurs at the Kalgan River junction, and there are discontinuous gullies further upstream. While not certain, the incision appears to have formed only a few years after clearing.

### 3.2. Takalarup Catchment Sediment Budget

In this catchment, the sources of sediment are hillslope sheet and rill erosion, gullying, incision of the main channel, and remobilization of PSA during the latter part of the post-European period. Sinks are colluvium, PSA, channel floor, and the alluvial fan at the mouth of the creek adjacent to the left bank of the Kalgan River.

The sediment budget (Table 2) is divided into the periods pre-1903, 1903–1955, and 1955 to 1997 (when all measurements were made). Yield for the period prior to 1903 was estimated at 1.6–3.1 t/year (0.1–0.2 $t/km^2$/year). The sheet and rill erosion component, when estimated using SOILOSS with the cover factor approximated by ground cover of grass found in the least disturbed mallee and open eucalypt forest in the area, and corrected by applying regression Equation (2), resulted in a delivery to the valley floors, when corrected for the amounts trapped at the edges of floodplains, of 2.7 $t/km^2$/year. The near-natural sediment delivery ratio (SDR) is therefore between 4 and 7%. The catchment was a sediment trapping system prior to clearing, a conclusion supported by the sedimentology and stratigraphy of valley floors.

**Table 2.** Sediment Budgets for the Takalarup catchment.

| Period | Category | Sources | | | | Stores and Yield | | | | |
|---|---|---|---|---|---|---|---|---|---|---|
| | | Sheet and Rill Erosion (t) | | Gullies and Channels (t) | PSA [2] (t) | Fan | Yield | | | SDR (%) |
| | | | | | | (t) | (t) | t/km$^2$/yr | |
| | | Connected | Unconnected [1] | | | | | | | |
| <1903 | | 42 | | 0 | 0 | 0 | | | 0.1–0.2 | 4–7 |
| 1903–1955 | Uncleared | 200 | 100 | | | | | | | |
| | Cleared | 940 | 1500 | | | | | | | |
| | Total | 1140 | 1600 | 47,300 | 12,110 | 7300 | 29,000 | 36 | | 58 |
| 1955–1997 | Uncleared | 200 | 100 | | | | | | | |
| | Cleared | 2800 | 2260 | 5230 [3] | | | | | | |
| | | | | 12,140 [4] | | | | | | |
| | Total | 3000 | 2360 | 17,370 | 2240 | 3650 | 14,500 | 22 | | 64 |
| 1903–1997 | | 4140 | 3960 | 64,670 | 14,350 | 10,950 | 43,500 | 30 | | 60 |

[1] This is also a store. [2] This includes an estimate of temporary storage in the active channel. [3] New. [4] Reworked PSA.

The area of clearing determined from the 1943 aerial photographs suggests that for the period of 1903–1955, only the main valley floor was cleared to a maximum distance of ~700 m from the channel.

The SDR ranges between 58 and 64% for the two time periods since 1903 and is 60% for the whole period since 1903. The yield has increased from its 'natural' state by a factor of 14–27 × 10³.

For the Takalarup catchment, the mean annual suspended sediment yield from Equation (3) is 41.5 t/year (2.7 t/km²/year). This is ~8% of the average estimated yield for 1955–1997 (Table 2) and cannot be explained by including the unmeasured bedload which is typically assumed to be ~10% of the total load. For subcatchments ≤36 km² (at most, about twice the area of the Takalarup catchment), specific yields vary between 0.2 and 50.3 t/km²/year. Therefore, the estimate for 1955–1997 (Table 2) is within the bounds of the measured loads for, at most, the last 10 years. However, because of very high variability between subcatchments, a more detailed comparison is not possible. Therefore, the estimated yields in Table 2 are adopted.

For the period 1903–1955, 95% of the total erosion (including the sheet and rill erosion products that did not reach the channel network) came from channel (and gully) erosion. This figure was only ~76% during 1955–1997 because clearing increased the amount of sheet and rill erosion. The yield and SDR are essentially the same for the two periods. These two parameters, in this case, are insensitive to the area of clearing because of a smaller storage area in PSA and the fan between 1955 and 1977, thereby offsetting the lower rate of gully and channel incision. For the period 1903 to 1997, channel (and gully) erosion contributed ~89% of the total erosion.

The $^{137}$Cs content of two samples from Takalarup Creek is 0.534 ± 0.338 Bq/kg and 0.200 ± 0.147 Bq/kg, both almost zero values. Between 1955 and 1997, the mean specific sheet and rill erosion rate was 3.7 t/km²/year. $^{137}$Cs was used to estimate sheet and rill erosion rates as high as 5000 t/km²/year in the area [38]. Near the study site, estimates are available of 4763 t/km²/year at Kendenup (compared with 75 t/km²/year from farm data from nearby areas, and [26] and 606 t/km²/year at a site north of the Porongorup Hills (70 t/km²/year estimated from nearby farm dam data).

### 3.3. Takalarup Catchment Phosphorus Loss

There is more TP and P(HCO₃) in the <75μm fraction in the SM and AF deposits, but P(Bio) is nearly constant in the two size fractions. In two sediment samples taken from the active channels, there is much more TP, P(HCO₃), and P(Bio) than in section 4 samples. This is presumably the result of particle sorting.

The calculated mean annual yield (of essentially particulate P) from the catchment (Table 2) combined with the result from the main channel (Table 3) suggests a mean annual loss to the Kalgan River since 1903 of 146 kg TP, 36 kg P(HCO₃), and 0.1 kg P(Bio). The mean annual TP yields from the farm dam catchments ranging from 0.1 to 0.7 kg/ha/year, with an average of 0.3 kg/ha/year.

**Table 3.** Phosphorus data for Takalarup Creek catchment.

| Locations | P (total) | | P (HCO₃) | | P (Bio) | |
|---|---|---|---|---|---|---|
| | (mg/kg) | | (mg/kg) | | (mg/kg) | |
| | <75 μm | >75 μm | <75 μm | >75 μm | <75 μm | >75 μm |
| Profile 4 SM | 130 | 58 | 6 | 2 | 0.04 | 0.04 |
| Profile 4 AF | 140 | 48 | <2 | <2 | 0.04 | 0.04 |
| Tributary Sediment | 360 | 64 | 29 | 7 | 0.07 | 0.04 |
| Main Channel | 330 | 200 | 87 | 44 | 0.15 | 0.12 |

### 3.4. Dingo Creek Catchment Sediment Budget

This catchment adjoins the Takalarup Creek catchment; shallow gullies have cut through a shallow, sandy, alluvial fill which, in the upper catchment, is set within a bedrock gorge that is ~20 m deep. Samphire-covered swamps along the main valley are common.

The catchment was uncleared until 1956 according to Brent Counsel, who farms in the upper part of the catchment, an observation confirmed by both the 1943 aerial photographs and by Sue and Jim Hunt, who saw the land cleared. The first clearing was of the eastern flatter land. The valley floor, some of the riparian zones, and rocky hills were cleared by the late 1960s. Saline seeps appeared in the late 1960s, and scalds began to form as vegetation died and sheet and rill erosion occurred. One natural saline seep is evident in the main valley on the 1943 aerial photographs.

Gullies formed in the tributary valleys in January 1982, according to Brent Counsel, because of a storm of about 180 mm. Salt scalds expanded rapidly in the 1980s, moving headwards from the low gradient valley floor to adjacent colluvial footslopes. This process has connected the hillslopes to channels across floodplains and samphire swamps by producing a continuous slope replacing the break-of-slope that previously existed between the hillslope and valley floor, and by reducing vegetation on both the footslopes and the floodplain, thereby reducing resistance to runoff and sediment transport. This change allows the products of sheet and rill erosion to reach the channels. Prior to salinization, this sediment would not have reached the channels.

The sediment budget (Table 4) was constructed using the methods applied to the Takalarup Creek catchment. Incision of the valley floors and salt scalding that connected hillslopes to channels occurred at about the same time. Therefore, the budget begins in 1982. Sheet and rill erosion was calculated for hillslopes connected to channels either directly or, if a floodplain or swamp intervenes, where scalding has established a connection. PSA is very small and is being gradually removed by salt scalding.

**Table 4.** Sediment Budget for Dingo Creek and Salt Creek catchments, with the budget for the Takalarup Creek catchment for comparison.

|  | Dingo Creek (1982–1997) | Salt Creek (1956–1997) | Takalarup Creek [1] (1903–1997) |
|---|---|---|---|
| SOURCES |  |  |  |
| Sheet and rill (connected) (t) | 21 | 830 | 4140 |
| Main channel incision (t) | 16 | 6005 | 61,226 |
| Gullying (t) | 507 | 305 | 3444 |
| Salt scalds (t) | 289 | 201 | 150 |
| Total (t) | 833 | 7341 | 68,960 |
| STORAGES |  |  |  |
| PSA (t) | 50 | 6400 | 350 |
| In-channel (t) | 53 | 150 | 340 |
| Fan (t) | 7 | - | 10,950 |
| Total (t) | 110 | 6550 | 25,640 |
| YIELD [2] (t) | 723 | 790 | 43,320 |
| (t/year) | 48 | 19 | 460 |
| (t/km$^2$/year) | 6 | 1.3 | 30 |

[1] Modified from data in Table 2—in-channel storage was not included in Table 2. [2] Differences between sources and storages.

The calculated yields lie within the range of measured suspended loads for the last 10 years (Equation (3)). The SDR for the Dingo Creek catchment is 87%, consistent with the very small quantity of storage in this relatively steep catchment (Table 1). Channel and gully erosion contributed 63% of the total erosion, recalling that the sheet and rill erosion component was calculated for the entire catchment because salt scalds connect all hillslopes to the channels. Sheet and rill erosion is very low at 0.2 t/km$^2$/year.

*3.5. Salt Creek Catchment Sediment Budget*

This catchment adjoins the Dingo Creek catchment. The shallow incision in the lower reaches of the catchment exposes layered alluvium of the SM type seen in the Takalarup catchment. The SM rests directly on gneiss bedrock, and there is no equivalent of the AF at Takalarup. Further upstream, in this long, low, gradient (Table 1) catchment, the valley fill is sandy without evidence of SM.

The upstream part of the Salt Creek catchment was cleared in the 1950s during the soldier settler period following World War II, according to Sue and Jim Hunt. There was no clearing visible on the 1943 aerial photographs. Mike Easton, who lives near the upstream part of the catchment and is attempting to reduce the rate of dryland salinization by planting trees and saltbush, noted that clearing happened between 1956 and the late 1970s. The most upstream part on Warburton's farm was cleared in the 1960s and salt appeared in the 1980s.

Prior to the 1970s, Salt Creek was barely visible as a channel, according to Sue and Jim Hunt. The channel deepened considerably after clearing began, not by a single runoff event but gradually. Since about 1987, the creek has enlarged again. Salt scalds began to form and migrated from valley floors to adjacent hillslopes, as in the Dingo Creek catchment during the late 1970s and early 1980s.

Clearing began in the catchment before the 1980s, in the downstream areas. The 1943 aerial photographs show a cleared area of ~2.3 km$^2$. Saline seeps, marked by evaporites, occurred on lower footslopes adjacent to the right bank of the main creek downstream of the cleared area. Some of these seeps may be natural but scalding began by 1943. By 1993, only ~1.1 km$^2$ was uncleared and salt scalds occurred further upstream along both the main channel and the major left bank tributary. The downstream scalds also expanded.

Connectivity between hillslopes and the channels developed around 1956, based on the memories of local inhabitants. Therefore, the sediment budget (Table 4) is estimated for 1956−1997.

The yield is again within the range of modern values (Equation (3)). Gully and channel erosion contributed ~86% of total erosion. The SDR is 11% in this relatively low gradient catchment. While salt scalds do not connect all hillslopes to channels, the total erosion, in this case, only includes sheet and rill erosion that is connected to channels either by scalding or by juxtaposition of hillslopes and channels without an intervening floodplain.

## 4. Discussion

Researchers [9] previously identified two end-member sediment (and inferred P) budget types in low-relief agricultural catchments, as follows:

A—sheet and rill erosion dominate sediment movement, and therefore fertilizer P, which is retained mostly in surface soil layers [39], can also be a significant source of stream particulate P. In this type, hillslope erosion may lag changes to erosive land uses, and valley-floor sedimentation absorbs most sediment and particulate P. The sediment (and particulate P) delivery ratio is low but can increase as overbank stores fill, allowing more transport in-stream.

B—gully and channel erosion dominate sediment input to streams, and therefore drainage density plays a key role. As the channel (and gully) network expands after land-use change, sediment yield increases to a peak and then stabilizes as the channel network reaches its maximum volume. Sediment (and particulate P) yield then decreases as the channel network walls and beds stabilize, and some gullies infill, although in some cases particulate P may stay high as soluble P is absorbed on particles, and as sediment yield falls soluble P may rise because there is less sediment to absorb P. Valley-floor sedimentation rates peak early, and then decrease as overbank stores fill and in-stream transport increases. The sediment (and particulate P) delivery ratio is higher than in type A, for the same size and type of catchment, and peaks twice: first as the drainage density is rapidly increasing but before valley-floor sedimentation peaks; and second, at or soon after the drainage density reaches its maximum. Fertilizer P attached to soil particles may not constitute

most of the stream P because hillslope erosion contributes little particulate P or sediment to streams. This of course depends on the relative concentration of P in topsoils and subsoils, and whether particulate P signatures directly represent sediment loss mechanisms or represent a recombination of disparate sources of sediment and soluble fertilizer P [2,5]. This thesis is supported by considering measurements of soluble and particulate P at the hillslope and large catchment scales in the Kalgan catchment. At the hillslope scale, dissolved P is estimated to represent 96–99% of the P lost, with transformation to 40% at the large catchment scale due to interaction of hillslope-derived dissolved P with bank- and gully-derived sediment [2].

The estimated P yield from the Takalarup catchment was about twice that determined from Equation (4), suggesting that about 50% of the yield comes from channel erosion rather than from the surface of hillslopes. The Takalarup catchment P yield would be about twice (260 kg TP) the channel erosion P loss estimates. However, increased TP concentrations in surface soils because of fertilization since clearing [39,40] indicates that this is probably an overestimate of the contribution of TP from channels and gullies. On average in the Kalgan River catchment, surface soils have twice the TP content as subsurface soils.

Examples of the sediment end-members are the Russian Plain for Type A and various catchments in southeastern Australia for Type B. Since 1696 AD, over the entire Russian Plain, $99 \times 10^9$ m$^3$ of soil was lost from hillslopes by sheet and rill erosion. In addition, $4 \times 10^9$ m$^3$ of soil was mobilized by gullying [41]. Gullying has contributed only 3.9% of total erosion, and 97% of the total was redeposited on the plain; that is, the SDR for the entire plain is only 3%. Large numbers of low-order stream channels are completely infilled by sediment so that drainage density declined despite gullying [42].

The almost zero $^{137}$Cs levels measured in the Takalarup Creek catchment sediment suggest little or no topsoil present compared to the 24% shown by the budget for 1955–1997. There are two possible explanations for these results. First, the figure of 24% refers to the entire period 1955–1997. The $^{137}$Cs values could reflect a recent period of very active channel and gully erosion relative to sheet and rill erosion. Phil Wishart observed that intense storms on 12, 13 December and 20–26 February 1996, produced a waterflow of ~1 m over the floodplain at the lower end of the creek, and severe channel erosion of a small head cut in the lower reach between ~1992/1993 and 1997. Second, the $^{137}$Cs radionuclide was depleted because it could not absorb particles as a result of either competition with cations in the saline water of the creek or cation exchange [43]. It is also possible that the initial labelling of the particles on the hillslope was inefficient because of the hydrophobic nature of the soils [44]. Whichever explanation holds, $^{137}$Cs cannot be used to test the calculated proportion of topsoil in the river sediments. The hydrophobic nature of the soils in this area implies that labelling was incomplete, and much of the nuclide was lost in runoff. Therefore, $^{137}$Cs concentrations in the soils are lower than they would be in non-hydrophobic soils and erosion is overestimated.

In southeastern Australia, only a small percent of the sediment being transported in a gully with a catchment of 5.1 km$^2$ is derived from sheet and rill erosion [22]. About 8% of the sediment in transport in gullies in the Jerrabomberra Creek catchment of 136 km$^2$ was derived by sheet and rill erosion during the period 1944 to 1991 [23]. Only 2–10% of sediment in transport in a 1.2 km$^2$ gullied catchment at Wirragulla came from sheet and rill erosion [24]. At a scale more comparable with the Russian Plain, >80% of suspended sediment in channels of the mid-Murrumbidgee River (area of 13,500 km$^2$) comes from channel banks and gully walls [29]. In all these cases, drainage density has increased since European settlement.

The three catchments in southwestern Australia are mostly examples of Type B, with the contribution to total erosion from channel and gully erosion ranging from 63 to 95%. The SDR varies between 8 and 64%, and drainage density (DD) between 0.6 and 1.2 km/km$^2$. The average valley floor gradient of the channelized part of each catchment varies from 0.0129 to 0.0256 (Table 1). There are no clear relationships between the proportion of gully

and channel erosion, SDR, drainage density, total channel length, or gradient. This may be a result of data being available for only three catchments.

The quantity of gully and channel erosion/year/km length of channel appears to be related to depth of alluvial fill in valley floors. Takalarup catchment has the deepest (incised) alluvial fill (≤3 m) and a gully/channel sediment production rate of 114 t/year/km length of channel. Dingo Creek and Salt Creek catchments have incised alluvial fills that, on average, are ≤ 1 m deep, and gully/channel sediment production rates of 105 and 371 t/year/km length of channel, respectively.

The response curve of sediment yield from disturbed catchments rises rapidly as incision begins, then falls to a new value that is usually higher than that of the pre-disturbed state [9,23,45]. The budget approximately resolved for different time periods in the Takalarup catchment (an example of Type B) provides additional information. Prior to 1903, when the native vegetation was intact, the yield was very low, and the only erosion was by sheet and rill processes. The SDR was also very low, so the catchment was a sediment trap (Figure 7).

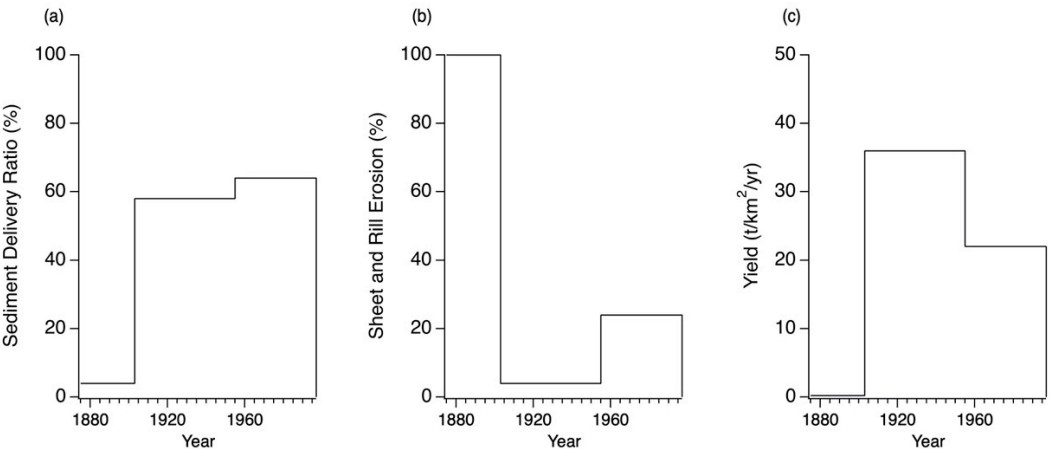

**Figure 7.** History of (**a**) SDR, (**b**) proportion of sheet and rill erosion, and (**c**) yield in the Takalarup Creek catchment.

It is noteworthy that SOILOSS overestimates the sediment yield in this study by a factor as high as 41. This could be partly the result of soil hydrophobicity [44], so that erodibility is much lower than indicated by soil texture.

During the period 1903 to 1955, when only ~9% of the catchment was cleared, mostly along the valley floor in the lower reaches of the catchment, major runoff in 1939 and again in 1955 produced incisions. This new source of sediment reduced the proportion of sheet and rill erosion to 5% and the yield increased by a factor between 180 and 360. Between 1955 and 1997, only ~7% of the catchment was uncleared, and the sheet and rill erosion contribution to total erosion rose to 24%, partly because of the clearing but also because most of the channel and gully incision had already occurred. However, the SDR and yield were essentially the same as during the period 1903 to 1955.

These results suggest that yield and SDR can be insensitive to major changes in the cleared area if changes to erosion and storage approximately balance. In the case of Takalarup, sheet and rill erosion increased from the earlier period (1903–1955) to the later period (1955–1997), but gully and channel erosion decreased. Therefore, total erosion decreased, but storage as PSA and in the fan also decreased.

In a study where temporal resolution of the sediment budget was superior to that achieved here, [46] found for the 360 km$^2$ the Coon Creek catchment in Wisconsin, yield remained constant between 1853 and 1993 despite a four-fold decrease in net upland sheet and rill erosion, and sediment deposition rates in the lower main valley were only 3–4% of the 1930s rates (the highest level on record) because the upper main valley became a sediment source. The SDR rose from 9 to 32% because overall storage lessened to a greater degree than erosion inputs. These results, from a catchment in which channel and

gully erosion has varied between 26–53% of total erosion in different time periods, suggest to [46] that measurements of sediment yield alone are of limited value as an indicator of the effectiveness of, for example, better land management. This conclusion adds to the argument for the utility of sediment budgets as a management tool [47].

Salt scalding plays a role in the study catchments by both increasing total erosion and by connecting hillslopes to channels across what would otherwise be intervening floodplains. The scalds are sheet-like, usually only 10–20 cm deep with a vertical headwall that retreats both across floodplains and up colluvial footslopes. Once they reach footslopes, sufficient gradient exists for rills to form that become conduits for the products of sheet and rill erosion to reach channels. Large runoff events also produce sheets of sediment that move across the scalds into channels, unimpeded by vegetation that would have grown on the floodplains before salinization. Therefore, dryland salinization and scalding increase the SDR of hillslope contributions to channels.

The small amount of data collected on P in the Takalarup catchment suggests that incision of the valley fills has contributed ~50% of the TP lost to the Kalgan River since 1903. The remainder has come from sheet and rill erosion of hillslopes. However, these results only refer to particulate P. A detailed study of a 5.9 km$^2$ agricultural catchment near Albany shows that before riparian restoration, the annual average yield was 11.4 t/km$^2$/year of suspended sediment. After riparian restoration, the yield fell to <1 t/km$^2$/year. Total P exports, both before and after riparian restoration, were between 0.02 and 0.06 t/km$^2$/year, and filterable reactive P (FRP) (similar to P(Bio)) was between 0.01 and 0.04 t/km$^2$/year [5]. These results suggest that P is most likely leached in a soluble form through the sandy soils which are low in Fe and therefore cannot adsorb more P after a few years of fertilizer application [15]. The FRP/TP ratio increased for the post restoration period for several reasons, possibly because riparian re-vegetation reduced the area of exposed channel banks that previously adsorbed leached P, in addition to reduced available suspended sediment for adsorption of soluble P in the stream for the post-restoration period. This latter explanation seems more plausible given that ~50% of the P transported in the pre-restoration period was in a particulate form, and if we assume from the Takalarup Creek catchment that ~50% of the P is derived from incision, the work of [5] would therefore suggest a ~25% reduction in P export, when no reduction between the pre- and post-restoration periods was seen.

In a comparison to the P characteristics of exposed channel banks and surface soils in the Kalgan catchment [2], the former contains half of the Total P of surface soils, they contain one fifth of the Colwell P, they are 5 times more P retentive, they are 1–2 orders of magnitude less saturated with P, and based on these characteristics they would contribute at least an order of magnitude less than Total P and bioavailable P to waterways than surface soils. Hence, it is much less likely that exposed channel banks would contribute P, and much more likely that they would retain soluble P fractions, as proposed by [2,5].

These results lead to other thoughts about catchment management to reduce sediment and P export to the Kalgan River and estuary. Clearly riparian re-vegetation will help to reduce sediment yield, but not necessarily P yield. Also, management of hillslopes to reduce dissolved and particulate P loads at the source is essential. This then also brings into question the interpretation of water quality data as representing the processes by which nutrients are lost.

## 5. Conclusions

The following major conclusions are drawn:

1. The three small agricultural catchments studied lie between the two end-member sediment budget types identified by [9] but mostly are of Type B where gully and channel erosion dominate.
2. The channel and gully erosion proportion of the total erosion appears to be controlled by the depth of alluvial fill and therefore the depth of incision. The incision volume is not a function of drainage density or gradient because the depth (and volume) of inci-

sion post-clearing varies between catchments as a result of topographic variation that enabled the thickest Holocene valley fill to accumulate in the Takalarup catchment.

3. While this explanation of the proportion of channel and gully erosion is plausible, it is based on little data. Further data collection from other catchments is essential.

4. Clearing of only ~9% of the Takalarup catchment along the lower reaches of the main valley and channel after 1903 allowed a storm in 1939 to incise valley fills almost all to bedrock. The time lag between first clearing and incision was 29 years at Takalarup. At Moorialup Creek it was ~23 years. West of Redmond it was 10 years, and at Noorabup only a few years. The smallest lag was at Noorabup, suggesting that decay of organic matter and roots, and destruction of soil structure, can occur within a few years. Any sizeable runoff event thereafter can produce incision.

5. Sediment yield and SDR at Takalarup have varied little since 1903 despite major changes of clearing, sheet and rill erosion rates, and storage of sediment. Yield and SDR, in this case, are insensitive to changes of vegetation cover within the catchments, like the case of Coon Creek in the U.S.A., because of adjustments to storage. This adds weight to the argument that sediment yield is sometimes a poor indicator of the state of a catchment, and of the effect of land use.

6. Dryland salinization following land clearing has both increased total erosion and connected some hillslopes to channels by scalding. This process increases the SDR of the hillslope to channel a sediment transport system.

7. The idea that particulate P yield of Takalarup Creek could be reduced by revegetation of riparian zones is unlikely because of leaching of soluble P through the sandy soils. Riparian management should reduce sediment yield, however.

8. The construction of the sediment budgets was made possible by field measurements combined with essential information from local farmers and residents.

**Author Contributions:** Conceptualization: R.J.W.; methodology: R.J.W.; software: R.J.W. and D.M.W.; validation: R.J.W. and D.M.W.; formal analysis: R.J.W. and D.M.W.; investigation: R.J.W. and D.M.W.; resources: R.J.W. and D.M.W.; data curation: R.J.W.; writing—original draft preparation: R.J.W.; writing—review and editing: R.J.W. and D.M.W.; visualization: R.J.W. and D.M.W.; supervision: R.J.W.; project administration: R.J.W.; funding acquisition, R.J.W. All authors have read and agreed to the published version of the manuscript.

**Funding:** This research was funded by Australian Research Council and the National Riparian Lands Program of Land and Water Australia.

**Institutional Review Board Statement:** Not applicable at the time the study was performed.

**Informed Consent Statement:** Not applicable at the time the interviews were done.

**Data Availability Statement:** The corresponding author can supply data.

**Acknowledgments:** Adrian Reed introduced the field sites for R.J.W. Local farmers and residents were generous with information and hospitality, particularly Phil Wishart, Ian Lock, Laurel Lock, Jack Rowe, Sue and Jim Hunt, Neil Heffernan, Mike Easton, Brent Counsel, Dudley Wise, and Peter Buxton. John Kinnear found copies of surveyors' field books and plans. Don McFarlane and Ruhi Ferdowsian provided advice on soil erosion and salinization respectively. Eri Leong provided expert research assistance.

**Conflicts of Interest:** The authors declare no conflict of interest.

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
