# Peer review of "Sediment Budgets for Small Salinized Agricultural Catchments in Southwest Australia and Implications for Phosphorus Transport"

_water, doi:10.3390/w13243564_

Round 1

Reviewer 1 Report

The study explored the sediment budget in the Kalgan River catchment, along with other rivers in southwestern Western Australia.

This article is well organized and structure. The Kalgan Catchment Environment has a complete and detailed description. The results and discussion are detailed and complete. Most of the Figures and Tables are clear.

Specific comments:

  1. The parameters Y and X in regress equations (1) and (3) use different units. Is it possible to use the same units?
  2. Sections 2.7 and 2.8 may modify to Section 3. Results (original 2.7) and 3.1 Takalarup Creek Catchment - Land Use, Erosion and Salinisation History (original 2.8).
  3. Table 2 the connected and unconnected may use symbols to distinguish.
  4. Figure 7 is not clear.

Author Response

Review 1

Open Review

English language and style

( ) Extensive editing of English language and style required  
( ) Moderate English changes required  
( ) English language and style are fine/minor spell check required  
(x) I don't feel qualified to judge about the English language and style  

Yes

Can be improved

Must be improved

Not applicable

Does the introduction provide sufficient background and include all relevant references?

(x)

( )

( )

( )

Is the research design appropriate?

(x)

( )

( )

( )

Are the methods adequately described?

( )

(x)

( )

( )

Are the results clearly presented?

( )

(x)

( )

( )

Are the conclusions supported by the results?

(x)

( )

( )

( )

Comments and Suggestions for Authors

Without details the comment above about the clarity of the presentation cannot be assessed and responded to. However some additional detail has been provided on methods.

The study explored the sediment budget in the Kalgan River catchment, along with other rivers in southwestern Western Australia.

This article is well organized and structure. The Kalgan Catchment Environment has a complete and detailed description. The results and discussion are detailed and complete. Most of the Figures and Tables are clear.

Thank you for these positive remarks.

Specific comments:

  1. The parameters Y and X in regress equations (1) and (3) use different units. Is it possible to use the same units? The parameters have different units and relate different phenomena. So, no change has been made.
  2. Sections 2.7 and 2.8 may modify to Section 3. Results (original 2.7) and 3.1 Takalarup Creek Catchment - Land Use, Erosion and Salinisation History (original 2.8). As far as we are aware, the adopted numbering follows the journal’s requirements.
  3. Table 2 the connected and unconnected may use symbols to distinguish. The two columns are clearly marked ‘connected’ and ‘unconnected’. No change has been made.
  1. Figure 7 is not clear. This figure has been updated to include left and right y axis titles and labelling to support interpretation. In addition, the figure caption has been edited to support interpretation.

Reviewer 2 Report

Detailed comments on Wasson and Weaver

Abstract:

The 1st sentence is weird English and not needed anyway

Line 30 Sedimentation is NOT a serious problem globally, certainly not in the way that eutrophication is. It depends where and how the sediment is deposited.

Line 32 remove more

Line 32 soluble P is the dominant form of P in sandy catchments because phosphate is not removed from solution by adsorption processes.  

Line 42 How common various types ..of what? Are – there are many places where the authors write text which assumes we know what they are referring to, without explanation.

Line 44 what has salinization got to do with soil erosion?

Line 45 what is a sediment budget type?

Line 46 temporal trajectories of change .. of what?

Line 47-48 This study wants to explore the role of dryland salinisation (which is generally defined as salt precipitation in surface soils) to soil erosion. I fail to understand why they should be connected.

It is very unclear form the introduction what is the problem being studied – soil erosion (which later on seems to be over very odd long time periods and salinisation which doesn’t seem to this reviewer to have any connection to soil erosion. Where does P come from? Is it natural? Is it from pollution and if so what sort of pollution?

Since the study seems to be mainly about erosion, why is sedimentation a problem?

Line 72 TSS is Total Suspended Solids and has no connection to highly saline groundwater.

Line 73-82 Gives the reader a description of the history of the catchment before European settlers and then through settlement. The text goes onto to use the testimony of old timers in the region as data in a sediment budget study. That is problematic.

Line 93-95 This is the materials and methods section and here the authors make a statement about P in rivers “40% of P measured in rivers is dissolved, and 60% is in particulate form [17}, implying that surface derived particulates are the main vector for P delivery.” This is a statement of interpretation of data not methods. Furthermore as P geochemist I would argue that the statement is not scientifically justified anyway, since dissolved P can be converted into particulate P by processes in the river.

Line 98 add ‘the salt content’ is rising

Where does this salt come from?

Section 2.2 What is the cause of salinisation, why is it increasing and what is being salinized?

Line 121 define what is the latter part of the first half of the nineteenth century? Why do we need to know as this study is about modern processes?

Streams are not salinized, soils are.

Line 152 The study refers to ‘other physical and chemical measurements in this study to support interpretation of sediment and P data’. But this is the methods section. What data and how was it measured?

Line 165 Why is organic content removed from mineral sediment since it was part of the original soil. Also how was organic matter determined? This is not explained though my guess is LOI was used which is very problematic if any CaCO3 is present.

Line 167 How was the material derived from dam walls corrected for stock trampling, wave attack and sheet and rill erosion.

Line 182 Sentence is not logical

Line 192 what is a salt scald

Line 198-203 How can you estimate soil erosion before dams were put in (and detailed maps existed) and why do you want to.

Line 212-215 This mentions Cs137 as a measure to estimate the proportion of topsoil in the river sediment. This is problematic in multiple ways. I start by pointing out that the authors do not explain how they measured 137Cs. This reviewers understanding is that 137Cs was a result of nuclear bomb testing which peaked in the early 1960’s and disappeared as a source with the end of atmospheric testing in 1963. There was of course some redistribution of 137Cs after that time so the signal did not disappear but the authors would need to show the input function and soil profiles to justify its use in this way.

The paragraph 216-221 is not justified for the reasons this reviewer has just written.

Line 222 what is PSA? If you don’t know what is PSA, then the rest of this paragraph does not make sense

Line 238-249 There is a lot of assuming here

Paragraph starting on line 2.6:

I am a P geochemist and these methods clearly come from soil science. Even with those comments, what is written is almost impossible to follow.

Kjeldahl is a digestion method designed to measure N in soils not P.

Reference [26] is the standard method to measure dissolved phosphate in seawater but is often used in other aquatic systems. It is not a method involving any sort of soil digestion.

Line 263 Phosphorus was extracted from what sort of sample with NaHCO3 and what does that extract measure.

Line 265 Why is phosphate adsorbed on iron oxide impregnated filter paper a measure of bioavailable P. It is some sort of measure of adsorbable phosphate.

At this point the reviewer gave up.  

Author Response

REVIEWER 2

English language and style

( ) Extensive editing of English language and style required  
( ) Moderate English changes required  
(x) English language and style are fine/minor spell check required  
( ) I don't feel qualified to judge about the English language and style  

Yes

Can be improved

Does the introduction provide sufficient background and include all relevant references?

( )

Is the research design appropriate?

( )

Are the methods adequately described?

( )

Are the results clearly presented?

( )

Are the conclusions supported by the results?

( )

Must be improved

Not applicable

(x)

( ) How can it be improved? Without suggestions we cannot respond

( )

(x)

( )

(x)

( )

(x)

( )

(x)

Comments and Suggestions for Authors

Detailed comments on Wasson and Weaver

Abstract:

The 1st sentence is weird English and not needed anyway It is not clear in what way this sentence is ‘weird’. Nonetheless, we have changed it to: Examples of sediment budgets are needed to document the range of budget types and their controls. Also, it is essential to one of the purposes of the paper.

Line 30 Sedimentation is NOT a serious problem globally, certainly not in the way that eutrophication is. It depends where and how the sediment is deposited. A debatable point and seeing there are no suggestions for change none has been made.

Line 32 remove more Done

Line 32 soluble P is the dominant form of P in sandy catchments because phosphate is not removed from solution by adsorption processes.  Text and supporting reference added.

Line 42 How common various types ..of what? Are – there are many places where the authors write text which assumes we know what they are referring to, without explanation. The reference here is to #9 which provides a complete explanation of this proposition. We cannot reproduce all this argument here. We expect readers to consult this reference if they wish to know more. As for the second sentence, no details are provided.

Line 44 what has salinization got to do with soil erosion? For greater clarify the sentence has been changed to: The study catchments examined here present an opportunity to explore a role in a sediment budget for dryland salinization, a process which follows clearing of native vegetation and also leads to the death of remaining vegetation or that which grows on parts of the cleared land, thereby reducing resistance to soil erosion.  

Line 45 what is a sediment budget type? This is explained later but reference to #9 has been repeated here.

Line 46 temporal trajectories of change .. of what? This is explained later and does not require elaboration here.

Line 47-48 This study wants to explore the role of dryland salinisation (which is generally defined as salt precipitation in surface soils) to soil erosion. I fail to understand why they should be connected. We have now added clarification (see above.)

It is very unclear form the introduction what is the problem being studied – soil erosion (which later on seems to be over very odd long time periods and salinisation which doesn’t seem to this reviewer to have any connection to soil erosion. Where does P come from? Is it natural? Is it from pollution and if so what sort of pollution? It is clear in our view that the study aims to construct sediment budgets where salinisation is an active causal process. This will also add to our understanding of budget types worldwide. We have explained how salinization removes vegetation and thereby reduces resistance to erosion. The sources of P are explained later.

Since the study seems to be mainly about erosion, why is sedimentation a problem? This is a paper about sediment budgets for whole catchments. Sediment budgets by definition include soil erosion, sediment storage, and sediment yield.

Line 72 TSS is Total Suspended Solids and has no connection to highly saline groundwater. Correct. This has now been changed to TDS.

Line 73-82 Gives the reader a description of the history of the catchment before European settlers and then through settlement. The text goes onto to use the testimony of old timers in the region as data in a sediment budget study. That is problematic. Why is it problematic? Oral history can be a valuable source of information in the absence of other records. We have used a method well known in human history of ‘triangulation’ whereby the testimony of one person is checked where possible against another person’s views. Without a detailed account of the problem from the perspective of the reviewer, we cannot respond.

Line 93-95 This is the materials and methods section and here the authors make a statement about P in rivers “40% of P measured in rivers is dissolved, and 60% is in particulate form [17}, implying that surface derived particulates are the main vector for P delivery.” This is a statement of interpretation of data not methods. Furthermore as P geochemist I would argue that the statement is not scientifically justified anyway, since dissolved P can be converted into particulate P by processes in the river. The text here has been modified for clarity and relevant references cited. When only measured at this scale, and without knowledge of the dynamics of P retention and release in streams [18,19], this data could be interpreted to mean that surface derived particulates are the main vector for P delivery.

Line 98 add ‘the salt content’ is rising Text has been clarified and additional reference added. It is more true to use the modified text than to say ‘the salt content’ is rising. Groundwater is highly saline (from cyclic salts), and water tables continue to rise due to reduced evapotranspiration from land clearing bringing inherently saline water closer to plant roots [13,20].

Where does this salt come from? Reference is now made to a cyclic source.

Section 2.2 What is the cause of salinisation, why is it increasing and what is being salinized? The cause is the removal of native vegetation which reduces evapotranspiration and leads to outcrop of the saline water table. This does not need to be explained as it is a well known process. But we have added a few extra words of explanation and the areas salinized have already been described.

Line 121 define what is the latter part of the first half of the nineteenth century? Why do we need to know as this study is about modern processes? It is not defined precisely but of course refers to the second quarter of that century. It is not defined because the exact timing is not known.

Streams are not salinized, soils are. Not so. Streams can be overwhelmed by salts just like soils.

Line 152 The study refers to ‘other physical and chemical measurements in this study to support interpretation of sediment and P data’. But this is the methods section. What data and how was it measured? Not all of the methods and measurements are in the Methods section because they differ between the study catchments. But they are all included.

Line 165 Why is organic content removed from mineral sediment since it was part of the original soil. Also how was organic matter determined? This is not explained though my guess is LOI was used which is very problematic if any CaCO3 is present. Because we are only interested in the mineral fraction of the sediment. Yes, LOI was used but there is no CaC03 in the soils. This sentence has now been changed to: Because only the mineral fraction is of interest, the volumes of mineral sediment were corrected for organic content by subtracting loss on ignition values.

Line 167 How was the material derived from dam walls corrected for stock trampling, wave attack and sheet and rill erosion. This sentence now reads: The volume, then mass after correction for bulk density, was also corrected for the material derived from dam walls by stock trampling, wave attack, and sheet and rill erosion by estimating the volume lost below a projected surface between those parts of the dam walls which were not disturbed.

Line 182 Sentence is not logical Without an explanation we cannot respond.

Line 192 what is a salt scald They had already been defined as follows: Salt scalds (shallow areas eroded after saline water discharge has killed vegetation)

Line 198-203 How can you estimate soil erosion before dams were put in (and detailed maps existed) and why do you want to. This is explained and was by using the RUSLE/SOILOSS model. It is necessary because part of the budgets is from before the construction of farm dams. No change has been made.

Line 212-215 This mentions Cs137 as a measure to estimate the proportion of topsoil in the river sediment. This is problematic in multiple ways. I start by pointing out that the authors do not explain how they measured 137Cs. This reviewers understanding is that 137Cs was a result of nuclear bomb testing which peaked in the early 1960’s and disappeared as a source with the end of atmospheric testing in 1963. There was of course some redistribution of 137Cs after that time so the signal did not disappear but the authors would need to show the input function and soil profiles to justify its use in this way. Text modified to: To test the calculated proportion of channel erosion products in the sediment actively being transported in the modern channel, 137Cs in the <10µm fraction was measured by high resolution gamma spectroscopy.  By the way, the isotope can still be found in soils worldwide, so ‘some redistribution’ is an understatement. But the reviewer is correct that had we gone further with this analysis then soil profiles would have been analysed for the input function.

The paragraph 216-221 is not justified for the reasons this reviewer has just written. Agree but we have not said otherwise.

Line 222 what is PSA? If you don’t know what is PSA, then the rest of this paragraph does not make sense This was defined in two paragraphs below equation 2 as; Post Settlement Alluvium (PSA).

Line 238-249 There is a lot of assuming here Without specific instances we cannot respond.

Paragraph starting on line 2.6:

I am a P geochemist and these methods clearly come from soil science. Even with those comments, what is written is almost impossible to follow.

Kjeldahl is a digestion method designed to measure N in soils not P. Not entirely true. The Soil Chemical Methods – Australasia refers to Kjeldahl P using sulphuric acid and sodium sulphate and a copper catalyst. Reference has been added for support

Reference [26] is the standard method to measure dissolved phosphate in seawater but is often used in other aquatic systems. It is not a method involving any sort of soil digestion. No response required as this reference simply details the analytical method of P determination following digestion

Line 263 Phosphorus was extracted from what sort of sample with NaHCO3 and what does that extract measure. Text modified for clarity. Plant available P was determined by extraction of soil samples in 0.5M NaHCO3 [31] and the inorganic P (P(HCO3)) was subsequently measured in the centrifugal extract colorimetrically as the phosphomolybdenum blue complex.

Line 265 Why is phosphate adsorbed on iron oxide impregnated filter paper a measure of bioavailable P. It is some sort of measure of adsorbable phosphate. A reference to iron oxide filter paper has now been included.

At this point the reviewer gave up.   Thanks for your help to this point.

Reviewer 3 Report

The relation between phosphorus transport and erosion is a known phenomenon and was investigated all over the world for many environments. This article does not describe very new knowledge but in a very precise way, shows the approach how to dealing with the cases where historical data are not so available. On the other hand, received results are very important for local communities and farmers. I am very impressed that the author has so good cooperation with the farmers and farmers know a lot about the history of their farms.  From a scientific point of view, I do not have any comments. I am only interested if it will be possible to add some more detailed information about SOILOSS equation and what is exact difference between it and RUSLE (I know I can to the references, but maybe it will be good to add one sentence)
Some editorial remarks:
Rows 54,59,115,118,155,446,451, 527,560,734. Please check the references. You have information about error in this rows.
Rows 61, 308, 326, 436 I think, here should be a reference, not a direct link.  
Row 80. Why do you underline the names?
Figure 6. No unit on vertical axes. 
Row 427. Please improve the chemical formula of sodium carbonate. 
Rows 492-496. Improve the font size
Row 645. “s” should be a small letter in Cs

Author Response

REVIEWER 3

Review Report Form 

Open Review

English language and style

( ) Extensive editing of English language and style required  
( ) Moderate English changes required  
( ) English language and style are fine/minor spell check required  
(x) I don't feel qualified to judge about the English language and style  

Yes

Can be improved

Must be improved

Not applicable

Does the introduction provide sufficient background and include all relevant references?

(x)

( )

( )

( )

Is the research design appropriate?

(x)

( )

( )

( )

Are the methods adequately described?

(x)

( )

( )

( )

Are the results clearly presented?

(x)

( )

( )

( )

Are the conclusions supported by the results?

(x)

( )

( )

( )

Comments and Suggestions for Authors

The relation between phosphorus transport and erosion is a known phenomenon and was investigated all over the world for many environments. This article does not describe very new knowledge but in a very precise way, shows the approach how to dealing with the cases where historical data are not so available. On the other hand, received results are very important for local communities and farmers. I am very impressed that the author has so good cooperation with the farmers and farmers know a lot about the history of their farms.  From a scientific point of view, I do not have any comments. I am only interested if it will be possible to add some more detailed information about SOILOSS equation and what is exact difference between it and RUSLE (I know I can to the references, but maybe it will be good to add one sentence). Thank you for these positive remarks. Reference 23 provides a detailed account of SOILOSS which is simply RUSLE calibrated using Australian data. The appropriate sentence has been modified and now reads: SOILOSS is a version of the RUSLE adapted for Australia by using locally measured data.

Some editorial remarks:
Rows 54,59,115,118,155,446,451, 527,560,734. Please check the references. You have information about error in this rows. The error statements have been removed because they were incorrect.
Rows 61, 308, 326, 436 I think, here should be a reference, not a direct link.   These are all websites so there is no reference other than the URLs.
Row 80. Why do you underline the names? This is standard practice for plant species names although they may be italicized by the editor.
Figure 6. No unit on vertical axes.  This has been corrected in the caption as we cannot change the drawing.
Row 427. Please improve the chemical formula of sodium carbonate. Done
Rows 492-496. Improve the font size The font size is the same throughout the manuscript.
Row 645. “s” should be a small letter in Cs Done

Round 2

Reviewer 1 Report

The authors had answered my question.

Author Response

Thank you for reviewing.